# Relaxed State-Adversarial Offline Reinforcement Learning: A Leap Towards Robust Model-Free Policies from Historical Data

## Abstract

Offline reinforcement learning (RL) targets the development of top-tier policies from historical data, eliminating the need for environmental interactions. While many prior studies have focused on model-based RL strategies, we present the Relaxed State-Adversarial Offline RL (RAORL), an innovative model-free offline RL solution. RAORL sidesteps model uncertainty issues by framing the problem within a state adversarial context, eliminating the need for explicit environmental modeling. Our method guarantees the policy's robustness and its capability to adapt to varying transition dynamics. Anchored in robust theoretical foundations, RAORL promises performance guarantees and presents a conservative value function that reflects average-case outcomes over an uncertainty set. Empirical evaluations on established offline RL benchmarks indicate that RAORL not only meets but frequently surpasses the performance of state-of-the-art methods.

## 1 Introduction

Reinforcement Learning (RL) is foundational for tackling sequential decision-making tasks. While online RL flourishes in simulations via direct environment engagement (Mnih et al., 2015; Silver et al., 2017), its transition to real-world scenarios often encounters logistical and financial problems during data collection. This becomes notably evident in critical areas such as healthcare and robotics. Conversely, offline RL presents a viable solution, using existing datasets to train policies without ongoing environmental interaction (Levine et al., 2020; Lange et al., 2012).

In RL, environmental interactions are vital for policies to investigate diverse states and assess action outcomes. However, in offline RL, where data is pre-gathered, there's a glaring obstacle: the dataset might not wholly represent the environment's intricacies. This leads to the potential discrepancy between the dataset's transition probabilities, $P_B(s'|s, a)$, and the true probabilities, $P(s'|s, a)$. Such deviations parallel the issues in robust RL, where there can be differences between simulated and real-world transition probabilities (Fujimoto et al., 2019). Given these parallels, we employ robust RL techniques to mitigate offline RL's intrinsic challenges.

Robust RL approaches address state transition probability deviations by finding a policy to perform best in the worst-case environment over a set of possible MDPs. It's logical to merge robust RL with model-based RL to tackle offline RL issues. Specifically, model-based RL methods develop an designed environment model from the dataset, allowing policy interaction during training. Despite recent advancements in model-based offline RL incorporating pessimistic dynamic models to handle model uncertainties (Rigter et al., 2022), these methods still face challenges when simulating stochastic environments (Antonoglou et al., 2022; Ozair et al., 2021). They can also introduce model errors and demand intricate hyperparameter tuning, raising questions about the reliability of synthetic samples (Van Hasselt et al., 2019; Lu et al., 2022; Yu et al., 2021). Conversely, applying online robust RL strategies to offline situations using model-free methods is complex. State perturbations might lead to out-of-distribution observations, resulting in exaggerated value function overestimations (Yang et al., 2022). This raises a critical question: *Can robust RL principles be effectively embedded within offline RL using a model-free approach?*

We present the *Relaxed State-Adversarial Offline Reinforcement Learning* (RAORL) algorithm, a novel approach to model-free offline RL, to answer the question. RAORL formulates the policy

challenge as a state-adversarial optimization problem (Lien et al., 2023), underpinned by a reward correction term that portrays an average-case scenario across an uncertainty set. Utilizing this relaxed state-adversarial optimization paradigm allows us to adeptly tackle the robust policy challenge, ensuring an offline, tractable optimization without online interactions. Accordingly, RAORL stands out for its ability to: 1) Measure and bridge the performance gap in real-world applications without online engagement, 2) Reduce dependency on precise transition model learning, and 3) Integrate seamlessly with established model-free offline RL methods like TD3+BC (Fujimoto & Gu, 2021) and ReBrac (Tarasov et al., 2023).

Through the D4RL benchmark (Fu et al., 2020), our evaluations validate RAORL's efficacy. It consistently surpasses baseline methods in various continuous-control tasks. With its empirical effectiveness and theoretical foundation, RAORL emerges as a top contender for risk-sensitive applications. We will release our code to the public upon the paper's acceptance.

## 2 RELATED WORK

### 2.1 MODEL-BASED OFFLINE REINFORCEMENT LEARNING

Model-based RL methods learn a model environment and subsequently generate synthetic data to optimize a policy. When training on synthetic data, they strive to enhance generalization (Ball et al., 2021; Wang et al., 2021). Since synthetic data may not be trustable, model-based methods typically employ uncertainty measures to regulate their model (Yang et al., 2021; Yu et al., 2020). For instance, MOReL (Kidambi et al., 2020) employs an ensemble of dynamics models to measure model uncertainty, yet the reliability of these estimates remains questionable. Meanwhile, COMBO (Yu et al., 2021), a method akin to CQL (Kumar et al., 2020), learns a Gaussian distribution over upcoming states and rewards via maximum log-likelihood. Despite most model-based offline RL approaches leveraging maximum likelihood estimates (Argenson & Dulac-Arnold, 2021; Matsushima et al., 2021), alternative strategies exist. They focus on model learning tailored for offline policy optimization, emphasizing accuracy under policy-induced state-action distributions (Lee et al., 2020; Rajeswaran et al., 2020; Hishinuma & Senda, 2021).

A study closely aligned with our work is (Rigter et al., 2022), which delves into the maximin formulation of offline RL. While their methodology is model-based, it unavoidably inherits the limitations of such a formulation. Particularly, model-based methods grapple with challenges in modeling stochastic environments, as emphasized by Antonoglou et al. (2022) and Ozair et al. (2021). The potential for additional model errors, intricate hyperparameter tuning, and concerns about synthetic sample authenticity further complicate the policy training (Van Hasselt et al., 2019; Lu et al., 2022; Yu et al., 2021). In contrast, our approach generates pessimistic synthetic transitions without relying on model environments. This model-free perspective offers a distinctive avenue for offline RL, sidestepping the inherent challenges and complexities associated with model-based methods.

### 2.2 MODEL-FREE OFFLINE REINFORCEMENT LEARNING

Model-free offline RL is unique because policies do not interact with environments during training. This domain has spawned several approaches, including policy constraint methods, importance sampling, regularization, and uncertainty estimation. Specifically, policy constraint techniques ensure that the learned policy closely aligns with the behavior policy derived from the dataset. They fall into two groups: direct (Fujimoto et al., 2019; Kostrikov et al., 2021; Wu et al., 2020) and implicit (Kumar et al., 2019; Fujimoto & Gu, 2021; Wang et al., 2020), depending on their use of a model to represent the behavior policy. Importance sampling methods in offline RL (Nachum et al., 2019; Zhang et al., 2020) re-weight the state-action distribution in the offline dataset. Regularization techniques (Kumar et al., 2020; Yu et al., 2021; Singh et al., 2020) refine the learned function by introducing penalty terms. Lastly, uncertainty-based methods (Agarwal et al., 2020) balance conservative and off-policy RL techniques based on the model's confidence level.

Most prevailing strategies focus on identifying out-of-distribution actions (Yang et al., 2022). However, these models tend to be overly conservative, resulting in a pronounced gap in the generalization capability of RL. Contrarily, our proposed RAORL methodology accentuates model-free transition uncertainty training. This approach seamlessly integrates into existing methods and paves the way for superior generalization capabilities.

### 2.3 ROBUST REINFORCEMENT LEARNING

The Robust MDP techniques aim to optimize rewards, especially under worst-case conditions where testing environments differ from training ones (Nilim & El Ghaoui, 2005; Iyengar, 2005; Wiesemann et al., 2013). As dimensionality rises, the intricacy of robust MDP intensifies due to the expanding search space. To address this issue, Tamar et al. (2014) pioneered a dynamic programming approximation, advancing the scalability of the robust MDP model. This was further enhanced by Roy et al. (2017) for nonlinear predictions, ensuring convergence to a localized minimum. Later, research by Wang & Zou (2021); Badrinath & Kalathil (2021) investigated convergence speeds when integrating function approximations under specific conditions. Derman et al. (2021) showed that regularized MDPs, designed to manage uncertain rewards, fall within the domain of robust MDPs. Their focus on regularized MDPs was influenced by the lesser computational demand than conventional robust MDP methods. Additionally, Clement & Kroer (2021) crafted efficient updates using gradient descent to tackle distributionally robust MDP, improving convergence speeds. However, despite these advancements, current model environments remain restrictive for real-world applications.

Our methodology resembles the relaxed state-adversarial policy optimization (RAPPO) (Lien et al., 2023), which was a robust RL method in online scenarios. We adapt RAPPO for offline contexts and present a novel formulation to account for deviations between offline datasets and their actual environments.

## 3 PRELIMINARIES

### 3.1 NOMINAL MDPs AND OFFLINE DATASET MDPs

**A nominal Markov Decision Process (MDP)** is defined by the tuple: $M = (S, A, P_0, R, \rho_0, \gamma)$, where $S$, $A$ represent the state and action spaces, reward function $R(s, a)$ lies within the interval $[-R_{\max}, R_{\max}]$, $P_0(s'|s, a)$ denotes the transition function, $\rho_0$ is the initial state distribution, $\gamma \in (0, 1)$ is the discount factor. We consider Markovian policies, $\pi \in \Pi$, which map each state to a distribution over actions. The value function, $V_M^\pi(s) = \mathbb{E}_{a_t \sim \pi, s_t \sim P_0}\left[\sum_{t=0}^{\infty} \gamma^t R(s_t, a_t)\right]$, represents the expected discounted return, and the return where policies starting from an initial state distribution can be written as $J_{\rho_0}(\pi, P_0) = \sum_{s \in S} \rho_0(s) V_M^\pi(s)$. In addition, the state-action value function is defined as $Q_M^\pi(s, a) = \mathbb{E}_{a' \sim \pi}\left[R(s, a) + \gamma \sum_{s'} P_0(s'|s, a) Q_M^\pi(s', a')\right]$.

**Within offline RL**, the objective centers around optimizing the policy via a static dataset $B = (s_i, a_i, r_i, s_i')_{i=1}^{|B|}$, which is sourced from a nominal MDP. Given the nominal MDP $M$ and initial values $Q(s, a)$, we define the MDP induced by the offline dataset, denoted as $M_B = (S \cup \{s_{\text{term}}\}, A, P_B, R, \rho_0^B, \gamma)$. This MDP retains the original state and action spaces of $M$ but consists of extra terminal states, $s_{\text{term}}$. In this context, the transition probabilities for $M_B$ are given by $P_B(s'|s, a) = \frac{N(s, a, s')}{\sum_{\tilde{s}'} N(s, a, \tilde{s}')}$, where $N(s, a, s')$ is the occurrences of the tuple $(s, a, s')$ within $B$. If a particular $(s, a)$ is absent from the dataset, implying $N(s, a, \tilde{s}') = 0$, then $P_B(s_{\text{term}}|s, a) = 1$. In this case, $r(s, a, s_{\text{term}})$ is aligned with the preliminary value $Q(s, a)$ (Fujimoto et al., 2019).

### 3.2 ROBUST REINFORCEMENT LEARNING

Robust RL addresses the challenges faced in traditional RL when the environment is uncertain. Unlike standard RL, robust RL aims to ensure good performance even in the worst-case scenario. This is achieved by learning robust policies resilient to variations in the environment's dynamics. The fundamental concept behind Robust RL is the uncertainty set $\mathcal{U}$, which encompasses all possible transition dynamics the agent might encounter. By optimizing the worst-case performance over $\mathcal{U}$, robust RL ensures that the policy will perform adequately, even if the environment behaves adversarially within the bounds defined by $\mathcal{U}$. Mathematically, the optimization problem for robust RL can be defined as: $\pi = \arg\max_\pi \min_{P \in \mathcal{U}} J_\rho(\pi, P)$, where $\pi$ is the robust optimal policy that maximizes the minimum value over all possible environments in $\mathcal{U}$.

## 4 METHOD

The challenges of Offline RL arise because pre-gathered datasets may not fully capture all environmental dynamics. Consequently, there exists a misalignment between the transition probabilities observed in the dataset, $P_B(s'|s,a)$, and the true transition probabilities, $P(s'|s,a)$. This situation bears resemblance to the dilemmas faced in robust RL, wherein transition probabilities diverge between simulated and real-world settings. Given these analogous challenges, we advocate for the incorporation of robust RL methodologies to address the issues in Offline RL.

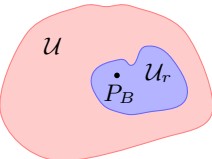

Figure 1: The diagram showcases the link between the risk-aware uncertainty set, $\mathcal{U}_r$, which envelopes the transition kernel $P_B$ of the offline dataset, and the comprehensive uncertainty set $\mathcal{U}$ encompassing all viable transition kernels. Clearly, $\mathcal{U}_r$ is a subset of $\mathcal{U}$.

The subsequent sections will navigate through the challenges of offline RL. Initially, we define the average deviation between every feasible real-world transition kernel $P \in \mathcal{U}$ and the transition kernel induced by the offline dataset $P_B$, and then explore *the performance lower bound on the offline dataset* in Section 4.1. Following this, Section 4.2 demonstrates how adopting a risk-aware policy – one that optimizes for the average scenario within an uncertainty set $\mathcal{U}_r$ determined from $P_B$, can improve the *performance lower bound*, although this predefined set may not perfectly match the real-world uncertainties. The relations between $\mathcal{U}, \mathcal{U}_r$, and $P_B$ is illustrated in Figure 1. In Section 4.3, we detail the use of the relaxed state-adversarial method to further increase the performance lower bound. This method aids in delineating the uncertainty set and optimizes the policy's average performance in a model-free context.

### 4.1 OFFLINE DATASET DEVIATION

To capture the uncertainty gap between reality and offline datasets, we consider the expectation of the offline dataset deviation by the following definition.

**Definition 1** (Expectation of Offline Dataset Deviation). *Given an offline dataset transition kernel $P_B$, we introduce a universal uncertainty set $\mathcal{U}$ to account for all feasible transition kernels in the real environment. This is mathematically captured by:*

$$\mathbb{E}_{P_0 \sim \mathcal{U}}[\mathbb{E}_{s,a} D_{TV}(P_0, P_B)] \leq \beta,$$

*where $\beta \geq 0$ and $D_{TV}$ denotes the Total Variation Distance.*

Consider the offline dataset MDP $M_B$ defined as $(S \cup \{s_{\text{term}}\}, A, P_B, R, \rho_0^B, \gamma)$ from which any policy $\pi$ is derived. Let $\mathcal{V}$ represent the set of unknown state-action pairs, such that $(s,a) \in \mathcal{V}$ if and only if $(s,a)$ is not present in the offline dataset. The term $T_{\mathcal{V}}^{\pi}$ represents the time taken to encounter these unknown states. With these definitions in place, we can present the offline dataset reality gap as follows:

**Lemma 1** (**Reality Gap:** Performance Gap between Offline Dataset and Reality (the universal uncertainty set)). *The value of any policy $\pi$ learned from $P_B$ on the universal uncertainty set $\mathcal{U}$ and the induced offline dataset transition kernel $P_B$ satisfies:*

$$\left| J_{\rho_0^B}(\pi, P_B) - \mathbb{E}_{P_0 \sim \mathcal{U}}(J_{\rho_0}(\pi, P_0)) \right| \leq \frac{2R_{max}}{1-\gamma} \mathbb{E}_{P_0 \sim \mathcal{U}}[D_{TV}(\rho_0, \rho_0^B)]$$

$$+ \frac{2\gamma R_{max}}{(1-\gamma)^2} \beta + \frac{2R_{max}}{1-\gamma} \mathbb{E}_{P_0 \sim \mathcal{U}} \mathbb{E}[\gamma^{T_{\mathcal{V}}^{\pi}}]. \quad (1)$$

The detailed proof can be found in Appendix A.1. Using Lemma 1, we can establish a lower bound on the performance of the offline dataset in relation to the optimal policy $\pi^*$:

**Theorem 1** (Offline Dataset Performance Lower Bound). *For any $\epsilon_\pi$ sub-optimal policy, we have:*

$$\mathbb{E}_{P_0 \sim \mathcal{U}}[J_{\rho_0}(\pi^*, P_0)] - \mathbb{E}_{P_0 \sim \mathcal{U}}(J_{\rho_0}(\pi, P_0)) \leq \epsilon_\pi + \frac{4R_{max}}{1-\gamma}\mathbb{E}_{P_0 \sim \mathcal{U}}[D_{TV}(\rho_0, \rho_0^B)] + \frac{4\gamma R_{max}}{(1-\gamma)^2}\beta$$
$$+ \frac{2R_{max}}{1-\gamma}\mathbb{E}_{P_0 \sim \mathcal{U}}\mathbb{E}[\gamma^{T_\nu^\pi}] + \frac{2R_{max}}{1-\gamma}\mathbb{E}_{P_0 \sim \mathcal{U}}\mathbb{E}[\gamma^{T_\nu^{\pi^*}}]. \quad (2)$$

The proof is in Appendix A.1. Theorem 1 highlights several challenges inherent in offline RL algorithms. First, the optimization error term, $\epsilon_\pi$, can be minimized by allocating more computational resources. Second, the distribution shift term $\beta$ represents the uncertainty in real-world dynamics. The other two terms are indicative of the offline dataset's comprehensiveness. Following these insights, we'll discuss how a risk-aware policy can enhance performance lower bound.

## 4.2 RISK-AWARE POLICY

Theorem 1 points out the challenges of applying offline dataset insights to real-world dynamics. Contrary to Robust RL strategies, which primarily target the worst-case scenarios, our approach focuses on a risk-aware policy concerning *an average case*. This policy operates over a designated risk-aware uncertainty set $\mathcal{U}_r$, where the expectation $\mathbb{E}_{P \sim \mathcal{U}_r}(\mathbb{E}_{s,a}D_{TV}(P, P_B))$ is constrained by $\leq \beta_r$. This can be formulated as: $\pi_r = \arg\max_\pi \mathbb{E}_{P \sim \mathcal{U}_r} J_\rho(\pi, P)$. *The subsequent Theorem 2 proves that leveraging a risk-aware policy enhances the policy's performance lower bound.*

**Theorem 2** (Risk-Aware Policy Performance Lower Bound). *For an $\epsilon_{\pi_r}$ sub-optimal risk-aware policy, we have:*

$$\mathbb{E}_{P_0 \sim \mathcal{U}}[J_{\rho_0}(\pi^*, P_0)] - \mathbb{E}_{P_0 \sim \mathcal{U}}[J_{\rho_0}(\pi_r, P_0)] \leq \epsilon_{\pi_r} + \frac{4R_{max}}{1-\gamma}\mathbb{E}_{P_0 \sim \mathcal{U}}[D_{TV}(\rho_0, \rho_0^B)]$$
$$+ \frac{4\gamma R_{max}}{(1-\gamma)^2}(\beta - \frac{1}{2}p_r\beta_r) + \frac{2R_{max}}{1-\gamma}\mathbb{E}_{P_0 \sim \mathcal{U}}\mathbb{E}[\gamma^{T_\nu^{\pi_r}}] + \frac{2R_{max}}{1-\gamma}\mathbb{E}_{P_0 \sim \mathcal{U}}\mathbb{E}[\gamma^{T_\nu^{\pi^*}}]. \quad (3)$$

The detailed proof is available in Appendix A.2. In Theorem 2, the component $p_r\beta_r$ signifies the reduced uncertainty associated with the risk-aware policy $\pi_r$. In essence, $p_r\beta_r$ captures the fraction of the total uncertainty $\beta$, addressed by this risk-aware policy. As a result, the term $\left(\beta - \frac{1}{2}p_r\beta_r\right)$ reflects the remaining uncertainty after implementing the risk-aware policy. Even though model-based strategies are prevalent in robust offline RL, mainly due to concerns related to out-of-distribution samples, we elaborate on a distinct model-free, risk-aware policy tailored for offline datasets in the subsequent sections.

## 4.3 MODEL-FREE RISK-AWARE POLICY IMPLEMENTATION

To obtain a more resilient model, we use the relaxed state-adversarial approach to account for uncertainties and potential adversarial situations in an offline dataset. Our decision to employ the surrogate perturbation method is primarily driven by two reasons: (1) it facilitates the generation of adversarial examples without necessitating an auxiliary estimated model, and (2) it is inherently suited for stochastic environments, a setting where model-based methods fall short (Antonoglou et al., 2022; Ozair et al., 2021). In essence, state-adversarial perturbation shifts current states towards neighboring states with minimal values. This shift is characterized by a state-adversarial transition kernel that bridges the standard MDP with the adversarial MDP. For clarity, let's define the $\sigma$-neighborhood of any state $s \in \mathcal{S}$ as $\mathcal{N}_\sigma(s) = \{s'|d(s, s') \leq \sigma\}$, where $d(s, s')$ is a distance metric. In our work, we employ the $L_\infty$-norm, denoted as $\|\cdot\|$.

**Definition 2** (Matrix of State Perturbations for Offline Dataset). *Consider an MDP characterized by the transition kernel $P_B$ derived from an offline dataset, a given policy $\pi$, and a perturbation measure $\sigma \geq 0$. For every state pair $i, j \in \mathcal{S}$, we define the matrix of state perturbations $Z_\sigma^\pi$ corresponding to $\pi$ as:*

$$Z_\sigma^\pi(i, j) = \begin{cases} 1, & \text{if } j = \underset{s \in \mathcal{N}_\sigma(i)}{\operatorname{argmin}} V^\pi(s|P_B), \\ 0, & \text{otherwise.} \end{cases} \quad (4)$$

The matrix identifies, for each state $i$, a neighboring state $j$ that has the lowest value function $V^\pi$, highlighting the least favorable outcomes of every state.

As noted by Lien et al. (2023), the $\arg\min$ in Equation 4 can be efficiently determined using the fast gradient sign method (FGSM) (Goodfellow et al., 2015) within continuous state domains. Given a value function $V$ characterized by parameter $\phi$, a state $s$, and a perturbation magnitude $\epsilon$, FGSM identifies the disturbed state $\Gamma(s) = s - \epsilon \times \text{sign}(\nabla_s V(\phi, s))$ with the lowest value. Here, $||s - \Gamma(s)|| \leq \epsilon$, and the gradient at $s$ is derived via back-propagation. Subsequently, the state value is iteratively updated using $V(s) = r(s, a) + \gamma V(\Gamma(s'))$. This approach eliminates the need to adjust the environment, contrasting with model-based algorithms.

**Definition 3** (Offline Dataset's State-Adversarial MDP). *Given a policy $\pi$, its associated state-adversarial MDP is characterized by the tuple $(\mathcal{S}, \mathcal{A}, P_\sigma^\pi, R, \mu, \gamma)$. The specific state-adversarial transition kernel for the offline dataset, $P_\sigma^\pi$, is expressed as*

$$P_\sigma^\pi(\cdot|s, a) = [Z_\sigma^\pi]^\top P_B(\cdot|s, a), \quad \forall (s, a) \in \mathcal{S} \times \mathcal{A}. \tag{5}$$

This transition kernel is biased towards the worst-case outcomes identified by the state perturbation matrix. Using the state-adversarial MDP $P_\epsilon^\pi$ usually helps improve the performance in the worst-case results (Kuang et al., 2022). However, setting too high a value for $\epsilon$ can result in overly cautious strategies (Lien et al., 2023). This emphasizes the importance of considering a spectrum of perturbation levels through the subsequent definition of the uncertainty set.

**Definition 4** (Offline Dataset's Uncertainty Set). *Given a perturbation radius $\epsilon > 0$, the uncertainty set of $P_B$ is defined as*

$$\mathcal{U}_\epsilon^\pi := \{P_\sigma^\pi : P_\sigma^\pi = [Z_\sigma^\pi]^\top P_B \text{ and } \sigma \leq \epsilon\}. \tag{6}$$

This uncertainty set captures all potential transition kernels under state-adversarial perturbations within the $\epsilon$ radius. The aim is to design a policy that remains robust against average-case scenarios within this set, which can be represented using the following relaxed state-adversarial transition kernel.

**Relaxed State-Adversarial Transition Kernel for Offline Dataset.** For given parameters $\epsilon > 0$ and $\alpha \in [0, 1]$, we define the $\alpha$-relaxed state-adversarial transition kernel as a weighted combination of the usual and state-adversarial transition kernels as:

$$P_\epsilon^{\pi,\alpha}(\cdot|s, a) = \alpha P_B(\cdot|s, a) + (1 - \alpha)P_\epsilon^\pi(\cdot|s, a). \tag{7}$$

Such a kernel achieves a deliberate balance, rendering the policy both suitable for real-world applications and resilient to unexpected disturbances. Subsequently, we demonstrate that $\alpha$ can be effectively interpreted as optimizing average-case scenarios within a relaxed state-adversarial transition kernel (Lien et al., 2023).

**Lemma 2** (Relaxation parameter $\alpha$ as a prior distribution $\mathcal{D}$ over uncertainty set $\mathcal{U}_\epsilon^\pi$). *For any distribution $\mathcal{D}$ over the state-adversarial uncertainty set $\mathcal{U}_\epsilon^\pi$, there must exist an $\alpha \in [0, 1]$ such that*

$$\mathbb{E}_{P \sim \mathcal{D}}[J(\pi|P)] = J(\pi|P_\epsilon^{\pi,\alpha}). \tag{8}$$

Emphasizing the significance of $\alpha$, its variations encapsulate unique prior assumptions. Specifically, modulating $\alpha$ allows us to represent a spectrum of distributions $\mathcal{D}$ and adapt policy training for multiple environments. Through the optimization of a relaxed state-adversarial policy, the performance lower bound is diminished as outlined in the subsequent theorem.

**Theorem 3** (Relaxed State-Adversarial Policy Performance Lower Bound). *For an $\epsilon_{\pi_{RA}}$ sub-optimal relaxed state-adversarial Policy policy, we have*

$$\mathbb{E}_{P_0 \sim \mathcal{U}}[J_{\rho_0}(\pi^*, P_0)] - \mathbb{E}_{P_0 \sim \mathcal{U}}(J_{\rho_0}(\pi_{RA}, P_0)) \leq \epsilon_{\pi_{RA}} + \frac{4R_{max}}{1 - \gamma}\mathbb{E}_{P_0 \sim \mathcal{U}}[D_{TV}(\rho_0, \rho_0^B)]$$

$$+ \frac{4\gamma R_{max}}{(1 - \gamma)^2}(\beta - \frac{1}{2}p_{RA}(1 - \alpha)) + \frac{2R_{max}}{1 - \gamma}\mathbb{E}_{P_0 \sim \mathcal{U}}\mathbb{E}[\gamma^{T_\nu^{\pi_{RA}}}] + \frac{2R_{max}}{1 - \gamma}\mathbb{E}_{P_0 \sim \mathcal{U}}\mathbb{E}[\gamma^{T_\nu^{\pi^*}}]. \tag{9}$$

The proof is in Appendix A.3. Within the framework of the state-adversarial uncertainty set, the term $p_{RA}(1 - \alpha)$ signifies the reduction in uncertainty achieved by the risk-aware policy $\pi_{RA}$. More explicitly, $p_{RA}(1 - \alpha)$ captures the portion of the overarching uncertainty, $\beta$, that is addressed by the risk-aware policy. As a result, the residual uncertainty, expressed as $(\beta - \frac{1}{2}p_{RA}(1 - \alpha))$, provides a measure of uncertainty that remains even after the risk-aware policy's intervention.

### 4.4 IMPLEMENTATION DETAILS

---

**Algorithm 1** Relaxed State-Adversarial Offline Reinforcement Learning (RAORL)

---

**Require:** Offline dataset $\{s_i, a_i, r_i, s'_i, d\}_{i=1}^N$, objective function $J$, step size parameter $\eta$, number of iterations $T$, number of update samples $T_{\text{upd}}$, uncertainty set radius $\epsilon$
1: Initialize policy $\pi_{\theta_0}$, value function $Q_{\phi_0}$
2: **for** $t = 0, \cdots, T - 1$ **do**
3:     Sample a tuple $\{s_i, a_i, r_i, s'_i\}_{i=1}^{T_{\text{upd}}}$ from the offline dataset
4:     Compute the corresponding state-adversarial transitions of the offline dataset batch $\{s_i, a_i, r_i, \arg\min_{s'_i \in \mathcal{N}_\epsilon(s'_i)} V^\pi(s'_i | P_B)\}_{i=1}^{T_{\text{upd}}}$ by Equation 5
5:     Compute the average scenario Bellman target $J(\pi_{\theta_t} | P^{\pi_{\theta_{t-1}}, \alpha})$ by Lemma 2
6:     Update value function $Q_{\phi_t}$
7:     $Q = \arg\min_Q \mathbb{E}_{(s,a,s') \sim D} \left[ \left( r(s, a) + \gamma \big( \alpha Q_{\bar{\phi}_t}(s', \pi(s')) + (1 - \alpha) Q_{\bar{\phi}_t}(adv(s'), \pi(s')) \big) - Q_{\phi_t}(s, a) \right)^2 \right]$,
8:     Update policy $\pi_{\theta_t}$
9:     $\pi = \arg\max_\pi \mathbb{E}_{(s,a) \sim D} \left[ \lambda Q(s, \pi(s)) - (\pi(s) - a)^2 \right]$
10: **end for**

---

Algorithm 1 details the presented method. During each iteration $t$, the update of policy $\pi_{\theta_t}$ can be achieved by using any off-the-shelf RL algorithm (e.g., TD3 (Fujimoto et al., 2018)) for optimizing the average-case return $J(\pi_{\theta_t} | P^{\pi_{\theta_{t-1}}, \alpha})$. We employ ReBrac (Tarasov et al., 2023) as our base algorithm, retaining its default hyper-parameters. For the relaxed state-adversarial component, we pick $\epsilon$ from the set $\{0.03, 0.05, 0.08, 0.1\}$ multiplied by state differences, which denote the absolute disparity between consecutive states. In addition, we determine $\alpha$ by choosing from the set $\{0.7, 0.8, 0.9\}$.

## 5 RESULTS AND EVALUATION

We conducted several experiments to evaluate the effectiveness of RAORL. Our objectives are three-fold: 1) *Performance Evaluation:* Comparing the proficiency of RAORL with prevailing state-of-the-art benchmarks, including model-based approaches: RAMBO (Rigter et al., 2022) and COMBO (Kumar et al., 2020)), and model-free approaches: S4RL(Sinha et al., 2022), ReBrac (Tarasov et al., 2023), ATAC (Cheng et al., 2022), IQL (Kostrikov et al., 2022), TD3+BC (Fujimoto & Gu, 2021), and CQL (Kumar et al., 2020); 2) *Ablation Study:* Understanding the impact of adversarial training on the algorithm's effectiveness; and 3) *Robustness Analysis:* Assessing the algorithm's stability under adversarial conditions. The evaluation spanned multiple environments:

**MuJoCo.** We conducted experiments on three distinct robotic environments (HalfCheetah, Hopper, Walker2D), each with three specific datasets ( Medium, Medium-Replay, Medium-Expert).

**AntMaze.** In this environment, the agent operates a robot with the objective of reaching a designated goal. Unlike MuJoCo, the reward system in AntMaze is sparse, rewarding the agent only upon successful goal attainment. The maze has three configurations (Umaze, Medium, Large), and the datasets vary (Fixed, Play, Diverse) based on the diversity in the starting points and goal locations used during data collection.

**Adroit.** This environment pertains to the control of a sophisticated 24-DoF simulated robotic hand. The tasks include hammering a nail, unlocking a door, spinning a pen, and grasping or relocating a ball. For each task, there are two distinct dataset types (cloned, and expert). The datasets are primarily human demonstrations focusing on tasks that demand precision in robotic manipulation.

### 5.1 PERFORMANCE EVALUATION

The experimental results outlined in Table 1 underscore the efficacy of our RAORL approach. While many previous methods demonstrated strong performances on the Mujoco datasets – a relatively simple environment – RAORL secured a marginally higher average reward than baseline techniques. As the environmental difficulty increased, leading methods such as RAMBO (Rigter et al., 2022),

| | RAORL | S4RL | RAMBO | ReBrac | ATAC | CQL | COMBO | TD3+BC | IQL |
|---|---|---|---|---|---|---|---|---|---|
| Halfcheetah-medium | 66.1 ± 1.2 | 48.6 | **77.6** | 65.5 | 53.3 | 44.4 | 54.2 | 42.8 | 47.4 |
| Halfcheetah-medium-replay | 51.0 ± 0.4 | 51.7 | **68.9** | 51.0 | 48.0 | 46.2 | 55.1 | 43.3 | 44.2 |
| Halfcheetah-medium-expert | **107.0 ± 3.2** | 78.1 | 93.7 | 101.0 | 94.8 | 62.4 | 90.0 | 97.9 | 86.7 |
| hopper-medium | **102.3 ± 0.4** | 81.3 | 92.8 | 102.0 | 85.6 | 86.6 | 94.9 | 99.5 | 66.3 |
| hopper-medium-replay | 100.4 ± 0.8 | 36.8 | 96.6 | 98.0 | **102.5** | 48.6 | 73.1 | 31.4 | 94.7 |
| hopper-medium-expert | 108.9 ± 4.2 | 117.9 | 83.3 | 107.0 | 111.9 | 111.0 | 111.1 | **112.2** | 109.6 |
| walker2d-medium | 86.8 ± 0.5 | 93.1 | 86.9 | 82.5 | **89.6** | 74.5 | 75.5 | 79.7 | 78.3 |
| walker2d-medium-replay | 85.0 ± 6.6 | 35.0 | 85.0 | 77.3 | **92.5** | 32.6 | 56.0 | 25.2 | 73.9 |
| walker2d-medium-expert | 112.2 ± 0.4 | 107.1 | 68.3 | 111.6 | **114.2** | 98.7 | 96.1 | 101.1 | 91.5 |
| Mujoco Average | **90.9** | 61.6 | 83.7 | 88.4 | 88.0 | 67.2 | 78.4 | 70.3 | 77.0 |
| | | | | | | | | | |
| pen-cloned | **106.6 ± 22.8** | 9.9 | 91.8 | 43.7 | 39.2 | - | 61.4 | 37.3 | |
| pen-expert | **154.9 ± 3.9** | - | 154.1 | 136.2 | 107.0 | - | 146.0 | - | |
| hammer-cloned | **3.7 ± 2.1** | 1.2 | 1.1 | 1.1 | 2.1 | - | 0.8 | 2.1 | |
| hammer-expert | **134.0 ± 0.6** | - | 133.8 | 126.9 | 86.7 | - | 117.0 | - | |
| door-cloned | 0.3 ± 0.4 | 0.5 | **6.7** | 3.7 | 0.4 | - | 0.1 | 1.6 | |
| door-expert | **104.2 ± 2.1** | - | **104.6** | 99.3 | 101.5 | - | 84.6 | - | |
| relocate-cloned | 0.4 ± 0.2 | -0.1 | **0.9** | 0.2 | -0.1 | - | -0.1 | -0.2 | |
| relocate-expert | **110.1 ± 0.7** | - | 106.6 | 99.4 | 95.0 | - | 107.3 | - | |
| Adroit Average | **76.5** | - | 74.9 | 51.6 | 43.6 | - | 64.6 | - | |
| | | | | | | | | | |
| Umaze | **97.5 ± 0.7** | 94.1 | 25.0 | **97.8** | - | 74.0 | 80.3 | 78.6 | 87.5 |
| Medium-Play | **91.5 ± 3.8** | 61.6 | 16.4 | 84.0 | - | 61.2 | 0.0 | 3.0 | 71.2 |
| Large-Play | **71.2 ± 16.5** | 25.1 | 0.0 | 60.4 | - | 15.8 | 0.0 | 0.0 | 39.6 |
| Umaze-Diverse | **87.7 ± 7.9** | 88.0 | 0.0 | **88.3** | - | 84.0 | 57.3 | 71.4 | 62.2 |
| Medium-Diverse | **86.7 ± 7.1** | 82.3 | 23.2 | 76.3 | - | 53.7 | 0.0 | 10.6 | 70.0 |
| Large-Diverse | **68.0 ± 7.6** | 26.2 | 2.4 | 54.4 | - | 14.9 | 0.0 | 0.2 | 47.5 |
| AntMaze Average | **83.8** | 62.8 | 11.2 | 76.8 | - | 50.6 | 22.9 | 27.3 | 63.0 |

Table 1: The performance of RAORL was benchmarked against baseline models, with results averaged across four random seeds. Following the work of (Fu et al., 2020), the scores in this table have been normalized using $(S_o - S_r)/(S_e - S_r)$, where $S_o$, $S_r$, and $S_e$ denote the rewards achieved by the offline policy, random policy, and expert policy. Note that the baseline results were copied from the papers of S4RL, RAMBO, ReBrac, and ATAC.

ATAC (Cheng et al., 2022), and TD3+BC (Fujimoto & Gu, 2021), which once dominated in certain Mujoco datasets, encountered a notable decline in their performance. It is worth noting that S4RL (Sinha et al., 2022) employs a comparable adversarial state training approach, propelling states towards their worst nearby states following transitions. However, such a direct application can yield excessively conservative outcomes in practical (Lien et al., 2023), especially real-world scenarios. This necessitates the adoption of a more tempered version of the state-adversarial technique. As illustrated in Table 1, RAORL demonstrated a marked superiority in complex environments like Adroit and AntMaze, noted as some of the most demanding in the D4RL benchmarks (Fu et al., 2020). Furthermore, while S4RL primarily offered an empirical analysis of state-adversarial methods, our research extends this by providing a theoretical foundation for the lower performance bound, thereby reinforcing the validity and effectiveness of employing state adversaries.

## 5.2 ABLATION STUDY

Given that RAORL is built upon ReBrac, we assessed the advantages of introducing a relaxed state adversarial approach to offline RL problems. According to Table 1, ReBrac serves as RAORL minus the relaxed state adversary. RAORL consistently outperformed or matched ReBrac across various environments, with the only exception being the *door-cloned* dataset, where no approach surpassed a score of 10. Given the extreme difficulty of this particular environment, the differences in scores between methods became less consequential. Excluding this special environment, RAORL demonstrated marked improvements over ReBrac, especially in datasets like *pen-cloned*, *AntMaze Medium-Play*, *AntMaze Large-Play*, *AntMaze Medium-Diverse*, and *AntMaze Large-Diverse*. Given the elevated challenge these datasets present compared to others, we deduced that incorporating relaxed state adversaries indeed enhances offline RL performance.

We also conducted experiments to determine if RAORL could enhance another foundational algorithm, namely Implicit Q-Learning (IQL). This was done to further substantiate its applicability and effectiveness. Table 2 (left) shows the results of IQL with and without the integration of the relaxed adversarial state technique. These experiments, run on three different seeds, show RAORL's notable improvements in performance.

|  | IQL | IQL+RA |
|---|---|---|
| Halfcheetah-medium-expert | 86.7 | **93.3 ± 1.5** |
| hopper-medium-expert | **109.6** | **112 ± 1.9** |
| walker2d-medium-expert | 91.5 | **112.4 ± 0.6** |

| Attack | RAORL | RORL-10 | RORL-2 |
|---|---|---|---|
| **0.025** | 72.1 | 75.8 | 48.1 |
| **0.05** | 61.2 | 65.0 | 33.2 |
| **0.075** | 53.1 | 53.5 | 32.6 |
| **0.1** | 41.7 | 44.3 | 29.1 |

Table 2: (Left) Experiments on IQL with and without our relaxed state adversarial technique. (Right) Robustness evaluation on Hopper-medium-expert over 4 seeds.

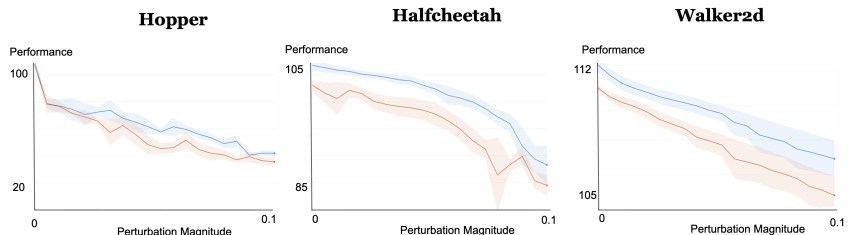

Figure 2: The blue and red solid lines depict the average performances of RAORL and ReBrac, in the presence of state perturbations. The vertical axis represents the normalized score, while the horizontal axis indicates the perturbation magnitude. The shaded areas illustrate the half standard deviation of the results, given that the experiments were conducted using four different seeds.

## 5.3 ROBUSTNESS ANALYSIS

In offline RL, a common and practical challenge arises when data collected from one system (machine A) is used to train an agent that will be deployed on a different but similar system (machine B). Even minor differences between these two machines can lead to distinct Markov Decision Processes (MDPs), posing a significant challenge in terms of MDP generalization. This situation underscores the importance of developing RL agents that can generalize effectively across varying MDPs. In essence, the agent must be capable of adapting to the nuances and potential discrepancies between the training environment (machine A) and the deployment environment (machine B). Therefore, we evaluated the robustness of policies trained using RAORL and ReBrac against adversarial perturbations in transition states. Specifically, in the evaluation, agents encountered different levels of adversarial perturbation based on their value functions. The perturbations were designed to transition the agent to states that minimize the expected return for actions taken from those states.

Figure 2 provides a side-by-side comparison under different perturbation levels for Medium-Expert datasets from the *Hopper*, *Halcheetah*, and *Walker2d* environments. The results highlight RAORL's superior resilience over the baseline, emphasizing the benefits of using relaxed state adversaries in offline RL contexts. Moreover, we compare RAORL with RORL(Yang et al., 2022) in robustness experiments because RORL is a model-free state-adversarial method that achieves state-of-the-art performance in the MuJoCo environment. As RORL is an ensemble-based model, we use *RORL-n* to indicate the use of $n$ ensembled models. Table 2 (right) demonstrates that our method achieves comparable results to RORL-10, which involves an ensemble size five times larger than ours. Note that RAORL notably outperformed RORL-2, where the two models have the same number of critics.

## 6 CONCLUSIONS

We have introduced RAORL, an innovative model-free strategy for offline RL that integrates state-adversarial perturbations, fostering robust policy development based on pre-collected datasets. Theoretically, RAORL offers a performance lower bound, showcasing resilience to discrepancies between the datasets and actual environments. Impressively, RAORL can effortlessly merge with existing model-free offline RL methods, further elevating policy performance. Empirical evaluations on widely recognized continuous control benchmarks underline its performance. In our studies, RAORL frequently outperformed leading methods, especially in complex tasks such as Adroit and AntMaz, demonstrating its effectiveness in offline RL applications.

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
