# OpenReview forum: "Relaxed State-Adversarial Offline Reinforcement Learning: A Leap Towards Robust Model-Free Policies from Historical Data"
_ICLR.cc/2024/Conference — Submitted to ICLR 2024_

### Official Review · Reviewer_6xeb · 2023-10-25

**Soundness:** 3 good
**Presentation:** 2 fair
**Contribution:** 3 good
**Rating:** 5
**Confidence:** 4

**Summary:**

This paper introduce a model-free offline RL approach, RAORL, by framing the problem within a state adversarial context. RAORL avoids model uncertainty issues and eliminates the need for explicit environmental modeling. The method ensures policy robustness, providing performance guarantees. Empirical evaluations demonstrate that RAORL consistently meets or surpasses the performance of existing SOTA methods.

**Strengths:**

* The studied problem model-free robust offline RL is important.
* The proposed method and the theoretical analysis seems reasonable.
* The empirical results on the D4RL benchmark and a robustness task shows the efficacy of the proposed method.

**Weaknesses:**

I appreciate the authors' efforts in investigating the studied problem; however, I think the current version requires several revisions and improvements.

* The methods section appears somewhat unclear. It is recommended that the authors include a subsection on the implementation of RAORL, detailing the learning objectives for both value functions and policy.

* RORL [1] is already a model-free state-adversarial method that achieves state-of-the-art performance on the MuJoCo benchmark and robust experiments. A thorough discussion and comparison with this work are necessary. Moreover, the proposed method seems quite similar to the adversarial state training in S4RL [2]. A more comprehensive discussion with closely related works is required to emphasize the significance of the method; otherwise, the novelty may be questioned.

* It appears that the method can be applied to different base algorithms besides ReBrac. Incorporating the method with approaches like ATAC and IQL could enhance the comprehensiveness of the experiments and support the claims made in the paper.

* In the state perturbation experiments, a comparison with the prior state adversarial baseline RORL is needed.

* The impact of each hyperparameter, $\epsilon$ and $\alpha$, on the performance of RAORL remains unclear.

* The authors should provide more implementation details in the Appendix to facilitate reproduction of the results.


[1] Yang, R., Bai, C., Ma, X., Wang, Z., Zhang, C., & Han, L. (2022). Rorl: Robust offline reinforcement learning via conservative smoothing. Advances in Neural Information Processing Systems, 35.

[2] Sinha S, Mandlekar A, Garg A. S4rl: Surprisingly simple self-supervision for offline reinforcement learning in robotics[C]//Conference on Robot Learning. PMLR, 2022.

**Questions:**

My questions are the same in the "Weakness" part.

* The authors should consider adding a subsection in the method section that details the learning objectives for both value functions and policy.

* A comprehensive discussion and comparison with RORL and S4RL are essential.

* Incorporating the method with approaches like ATAC and IQL could improve the comprehensiveness of the experiments.

* A comparison with the prior state adversarial baseline RORL is necessary in the state perturbation experiments.

* A hyperparameter study about the $\epsilon$ and $\alpha$ would be beneficial.

* To facilitate the reproduction of results, the authors should provide more implementation details in the Appendix.

---

> ### Author Response · Authors · 2023-11-21
> **Rebuttal**
>
> We thank reviewer 6xeb for the insightful comments. Below, we address the questions raised by the reviewer. We hope the replies could help the reviewer further recognize our contributions. Thank you.
>
> > Q: It appears that the method can be applied to different base algorithms besides ReBrac. Incorporating the method with approaches like ATAC and IQL could enhance the comprehensiveness of the experiments and support the claims made in the paper.
>
> A: We are currently conducting experiments to test our method with different base algorithm IQL, to further validate its applicability and effectiveness. The following table are results from integrating the Relaxed Adversarial State technique with IQL are promising. These experiments, run on three different seeds, show notable improvements in performance.
>
> |                          | IQL            | IQL+RA         |
> |--------------------------|----------------|----------------|
> | Halfcheetah-medium-expert| 86.7           | **93.3 ± 1.5**     |
> | hopper-medium-expert     | 109.6          | **112 ± 1.9**      |
> | walker2d-medium-expert   | 91.5           | **112.4 ± 0.6**    |
>
>
> > Q: In the state perturbation experiments, a comparison with the prior state adversarial baseline RORL is needed.
>
> A: Our methodology is primarily based on the ReBrac approach, making the performance comparison between our method and RORL particularly crucial. RORL utilizes an ensemble-based model, with "RORL-n" indicating the use of n ensemble models. In the following analysis on Hopper-medium-expert over 4 seeds, we demonstrate that our method achieves comparable results to RORL-10, which involves an ensemble size five times larger than ours. Furthermore, our approach notably outperforms RORL-2, which has the same numbers of critics of our model.
>
> | Attack magnitude | RAORL | ReBrac | RORL-10 | RORL-2 |
> |--------|-------|--------|---------|--------|
> | 0.025  | 72.1  | 67.1   | 75.8    | 48.1   |
> | 0.05   | 61.2  | 40.2   | 65.0    | 33.2   |
> | 0.075  | 53.1  | 35.3   | 53.5    | 32.6   |
> | 0.1    | 41.7  | 30.1   | 44.3    | 29.1   |
>
> > Q: The impact of each hyperparameter,  and , on the performance of RAORL remains unclear.
>
> A: In RAORL, we introduce two critical hyperparameters: the radius of the uncertainty set and the relaxed parameter alpha. The radius of the uncertainty set determines its size, directly influencing the range of potential state-action pairs considered during policy training. The relaxed parameter alpha, on the other hand, plays a pivotal role in defining the distribution over the uncertainty set. Varied values of alpha represent different prior assumptions about the environment. For instance, setting alpha to 1 suggests a belief that the nominal MDP is the most likely scenario, indicating confidence in the representativeness of the training data. Conversely, an alpha value of 0 implies a lean towards the worst-case MDP, preparing the policy for the most challenging possible scenarios.

---

> ### Author Response · Authors · 2023-11-21
> **Rebuttal**
>
> We thank reviewer 6xeb for the insightful comments. Below, we address the questions raised by the reviewer. We hope the replies could help the reviewer further recognize our contributions. Thank you.
>
>
> > Q: The methods section appears somewhat unclear. It is recommended that the authors include a subsection on the implementation of RAORL, detailing the learning objectives for both value functions and policy.
>
> A: Our algorithm can be found in algorithm 1. Specifically, inspired by TD3+BC (Fujimoto & Gu, 2021) and ReBrac (Tarasov et al., 2023), Objective function for the policy:
>
> $$
> \pi = argmax_{\pi} \mathbb{E}_{(s,a) \sim D} \left[ \lambda Q(s, \pi(s)) - (\pi(s) - a)^2 \right]
> $$
>
> The default values of the hyper-parameters($\lambda$) were used in the experiments (same as ReBrac (Tarasov et al., 2023)).
>
> Objective function for the average scenario value function is followed the lemma2:
>
> $$
> L_{TD}(\theta) = E_{(s,a,s') \sim D} \left[ \left( r(s,a) + \gamma \big( \alpha Q_{\bar{\theta}}(s', \pi(s’)) + (1-\alpha) Q_{\bar{\theta}}(adv(s'), \pi(s’)) \big) - Q_{\theta}(s, a) \right)^2 \right],
> $$
>
> We set perturbation radius from the set {0.03, 0.05, 0.08, 0.1} multiplied by state differences (in Section 4.4). The setting of attack budgets were chosen to reflect the mean magnitude of the state affected by actions taken in each environment.
>
>
>
>
> > Q: A thorough discussion and comparison with this work are necessary. Moreover, the proposed method seems quite similar to the adversarial state training in S4RL [2]. A more comprehensive discussion with closely related works is required to emphasize the significance of the method; otherwise, the novelty may be questioned.
>
> A: The main contribution of RAORL over S4RL is a theoretical guarantee for the lower bound of performance, further justifying the use of state-adversaries . Besides, we acknowledge the importance and “relaxed” technique beyond state-adversarial technique. In S4RL, the state-adversarial technique was identified as the most effective augmentation method based on their experiments. However, applying the state-adversarial technique in a straightforward manner can lead to overly cautious results (Lien et al., 2023). This is primarily because the state-adversarial technique inherently prepares for the worst-case scenario. Therefore, in practical applications, particularly in real-world problems, it becomes essential to adopt a "relaxed" version of the state-adversarial technique. The following table shows a comparative study of RAORL and S4RL in various experimental environments.
>
> |                          | RAORL         | S4RL           |
> |--------------------------|---------------|----------------|
> | Halfcheetah-medium       | **66.1**          | 48.6           |
> | Halfcheetah-medium-replay| **51.0**          | **51.7**         |
> | Halfcheetah-medium-expert| **107.0**        | 78.1           |
> | hopper-medium            | **102.3**         | 81.3           |
> | hopper-medium-replay     | **100.4**        | 36.8           |
> | hopper-medium-expert     | 108.9         | **117.9**         |
> | walker2d-medium          | 86.8          | **93.1**           |
> | walker2d-medium-replay   | **85.0**          | 35.0           |
> | walker2d-medium-expert   | **112.2**        | 107.1          |
> | Average                  | **90.9**          | 61.6           |
>
>
> |                       | RAORL         | S4RL          |
> |-----------------------|---------------|---------------|
> | antmaze-umaze         | **97.5**      | 94.1          |
> | antmaze-umaze-diverse | 87.7          | **88.0**      |
> | antmaze-medium-play   | **91.5**      | 61.6          |
> | antmaze-medium-diverse| **86.7**      | 82.3          |
> | antmaze-large-play    | **71.2**      | 25.1          |
> | antmaze-large-diverse | **68**        | 26.2          |
> | Average               | **83.8**      | 62.8          |

---

> > ### Comment · Reviewer_6xeb · 2023-11-22
> > **Thanks for your response**
> >
> > Thanks for your response. After carefully reading the response, my concerns persist. First, the paper was not updated. The discussion for the learning objectives, discussions about RORL and S4RL, and new results are not included in the revision. Moreover, the paper continues to lack the necessary implementation details that would enable the replication of the results, along with comprehensive studies on the introduced hyperparameters. Therefore, I will keep my current rating.

---

> > > ### Author Response · Authors · 2023-11-23
> > > **Response to Reviewer 6xeb**
> > >
> > > We thank the reviewer for the reply.
> > >
> > > **We uploaded the revived paper, and summarized the changes:**
> > > 1. Add S4RL, a state-adversarial method, comparison into Table 1.
> > > 2. Add Table 2 for incorporating our method to another baseline IQL.
> > > 3. Add Table 2 for the comparison to the state-of-the-art state-adversaria robust model.
> > > 4. Add the reason why reducing reality-gap (5.3 experiments) is especially important for offline RL.
> > > 5. Add objective functions of value function and policy in Algorithm 1.
> > > 6. Present Lemma 1 in absolute value form.
> > >
> > > **Implementation details on hyperparameters:**
> > >
> > > Regarding the implementation details, the baselines and our method were implemented on the ReBrac (Tarasov et al., 2023). The default values of the hyper-parameters were used in the experiment. We set perturbation radius from the set {0.03, 0.05, 0.08, 0.1} multiplied by state differences (in Section 4.4). The setting of attack budgets were chosen to reflect the mean magnitude of the state affected by actions taken in each environment.  Please refer to section 4.4 in our main paper. Moreover, we have submitted our code for reproduction.
> > >
> > > **Implementation details on Robustness Comparison (Section 5.3):**
> > >
> > > We defined the perturbation magnitude linearly with 20 points in the range $[0, 0.1]$. A perturbation of 0 signifies no perturbation, while a perturbation of 0.1 implies a perturbation strength of $0.1 \times | \text{ActualNextState} -  \text{State}|$, thereby ensuring that the norm of the difference between the actual state and the perturbed state is at most $0.1 \times  | \text{ActualNextState} -  \text{State}|$.
> > >
> > > As depicted in Figure 2, we analyzed the effect of different perturbation magnitudes on agent performance. The results demonstrate that RAORL maintains robust performance even under significant state perturbations.

---

### Official Review · Reviewer_K6G5 · 2023-10-26

**Soundness:** 2 fair
**Presentation:** 2 fair
**Contribution:** 2 fair
**Rating:** 3
**Confidence:** 3

**Summary:**

This paper introduces the Relaxed State-Adversarial Offline Reinforcement Learning (RAORL) method, a model-free offline RL approach that deals with model uncertainty issues by using a state adversarial context. By eliminating the requirement for explicit environmental modeling, RAORL offers a robust and adaptive policy formulation. It addresses challenges related to offline RL's dataset limitations and ensures performance in real-world applications without online interactions.

**Strengths:**

The primary contribution of this paper is the introduction of a robust offline reinforcement learning method that offers a lower bound guarantee. This aspect distinguishes the work and highlights its potential significance in the field of reinforcement learning.

**Weaknesses:**

1. The quality of writing in this paper could benefit from further refinement.  Some explanations are unclear.  For instance, in Lemma 1, formula 1 and formula 2 look identical, I had thought that formula 1 might have been inadvertently repeated.  But it turns out there are slight differences. I think there should be a better presentation such as using the absolute difference is bounded. Furthermore, the content of Lemma 1 raises questions for me.  The textual description seems to discuss the relationship of the policy's value learned from two different properties of P_B.  However, the formula appears to depict the policy's value learned from U and P_B.  I hope the authors recognize this distinction.  One refers to the value function learned under a specific P_B, while the other pertains to the average value function learned on U.

2. While this paper offers a plethora of proofs and new definitions, it lacks a coherent logical thread to effectively connect them. Additionally, the absence of adequate textual explanations for the theorems and lemmas makes the paper hard to follow and comprehend. There is no formal problem formulation for the problem the paper wants to solve. A more structured approach and clear expositions would greatly enhance its readability.

3. This paper introduces the concept of the Relaxed State-Adversarial Transition Kernel. However, the name doesn't seem to genuinely capture the essence of this transition kernel. I'm uncertain where the "State-Adversarial" aspect is present. The approach of crafting a new transition kernel using a linear combination appears to be already employed by many robust RL under model uncertainty methods, such as the R-contamination uncertainty set. You may check this paper "Policy Gradient Method For Robust Reinforcement Learning".

**Questions:**

Please see the Weaknesses, I will decide my final score after rebuttal.

**Details Of Ethics Concerns:**

Non.

---

> ### Author Response · Authors · 2023-11-21
> **Rebuttal**
>
> We thank reviewer K6G5 for the insightful comments. Below, we address the questions raised by the reviewer. We hope the replies could help the reviewer further recognize our contributions. Thank you.
>
> > Q: The quality of writing in this paper could benefit from further refinement. Some explanations are unclear. For instance, in Lemma 1, formula 1 and formula 2 look identical, I had thought that formula 1 might have been inadvertently repeated. But it turns out there are slight differences. I think there should be a better presentation such as using the absolute difference is bounded. Furthermore, the content of Lemma 1 raises questions for me. The textual description seems to discuss the relationship of the policy's value learned from two different properties of P_B. However, the formula appears to depict the policy's value learned from U and P_B. I hope the authors recognize this distinction. One refers to the value function learned under a specific P_B, while the other pertains to the average value function learned on U.
>
> A: We revise Lemma 1 in absolute difference and clarify the details in Lemma 1 as follows.
>
> 1. Relevance of U and P_B: As shown in Figure 1,  U represents the set of all possible real-world transition kernels, encompassing a wide range of scenarios and environmental dynamics. In contrast, P_B is specific to the offline dataset and represents the transitions recorded in that particular dataset.
>
> 2. Objective of Lemma 1: The lemma intends to quantify the performance gap between a policy learned exclusively from the offline dataset (P_B) and the policy evaluated against the universal set of transition kernels (U). This gap highlights the potential discrepancies and challenges in applying policies learned in a limited offline context to a broader range of real-world scenarios.
>
> 3. Lemma 1 in absolute difference :  \begin{align}
> \left| J_{\rho_0^B}(\pi, P_B) -E_{P_0 \sim U}(J_{\rho_0}(\pi, P_0)) \right| \leq & \frac{2R_{\text{max}}}{1-\gamma} E_{P_0 \sim U}[D_{\text{TV}}(\rho_0,\rho_0^B)]  + \frac{2 \gamma R_{\text{max}}}{(1-\gamma)^2} \beta + \frac{2R_{\text{max}}}{1-\gamma} E_{P_0 \sim U}{E}[\gamma^{\text{T}_{V}^\pi}].
> \end{align}
>
>
> > Q: While this paper offers a plethora of proofs and new definitions, it lacks a coherent logical thread to effectively connect them. Additionally, the absence of adequate textual explanations for the theorems and lemmas makes the paper hard to follow and comprehend. There is no formal problem formulation for the problem the paper wants to solve. A more structured approach and clear expositions would greatly enhance its readability.
>
> A: In the second paragraph of Section 4, we provided a logical thread of sections of 4.1, 4.2 and 4.3. In Section 4.1, we systematically measure the performance discrepancies between the policy shaped solely from the offline dataset ($P_B$) and its application in the universal set of transition kernels (U). Sections 4.2 and 4.3 further build upon this by demonstrating methods to enhance policy performance.
>
> > Q: This paper introduces the concept of the Relaxed State-Adversarial Transition Kernel. However, the name doesn't seem to genuinely capture the essence of this transition kernel. I'm uncertain where the "State-Adversarial" aspect is present. The approach of crafting a new transition kernel using a linear combination appears to be already employed by many robust RL under model uncertainty methods, such as the R-contamination uncertainty set. You may check this paper "Policy Gradient Method For Robust Reinforcement Learning".
>
> A: The term "State-Adversarial" in our Relaxed State-Adversarial Transition Kernel is defined in Definition 2 of our paper. It refers to the process where state transitions are adversarially perturbed towards those with lower value estimates, a concept critical for robustness in continuous state-action spaces. Our kernel thus represents a bridge between standard and adversarial MDPs, a novel contribution particularly tailored for continuous environments.
> While the "Relaxed" aspect of our kernel does resemble approaches like the R-contamination uncertainty set found in robust RL literature, the "State-Adversarial" facet specifically addresses continuous domains. This differentiates our work from that in "Policy Gradient Method For Robust Reinforcement Learning," which primarily addresses discrete spaces (as their algorithm requires the summation over all possible states and actions) and does not extend to the continuous settings that our paper focuses on.

---

> > ### Comment · Reviewer_K6G5 · 2023-11-22
> > **Thanks for the response**
> >
> > Thanks for the response!  I have one followup question: if state-adversarial means the process where state transitions are adversarially perturbed towards those with lower value estimates, shouldn't it be transition-adversarial?

---

> > > ### Author Response · Authors · 2023-11-23
> > > **Response to Reviewer K6G5**
> > >
> > > We thank the reviewer for the questions!
> > >
> > > Yes, state-adversarial is a transition-adversarial method. Hence the goal of RAORL is to be robust against transitions. Specifically, the goal of our study is to address two pivotal questions:
> > >
> > > **Q1: To what extent and through what mechanisms can we reduce the performance gap between offline datasets and real-world scenarios?**
> > >
> > > A1:
> > >
> > > Our primary aim is to lessen the performance gap typically observed when a policy developed using offline datasets (denoted as $P_B$) is applied in real-world scenarios, characterized by a diverse set of transition kernels (represented by $U$). We approach this in two key ways:
> > > 1. **Risk-Aware Policy Training:** We demonstrate that training a risk-aware policy over a carefully constructed uncertainty set $U_r$ can reduce the gap between its performance in training and real-world applications. This reduction is quantified by: $\frac{4 \gamma R_{\text{max}}}{(1-\gamma)^2} (\beta - \frac{1}{2}p_r \beta_r)$ (details in Theorem 2).This formula represents a theoretical measure of the improvement we can expect when using this risk-aware approach.
> > > 2. **Constructing a Practical Uncertainty Set:** We create an uncertainty set, $U_r$, using a state-adversarial uncertainty set  $U_\epsilon^{\pi}$ as outlined in Definition 4. The theoretical benefit of this approach is captured by $\frac{4 \gamma R_{\text{max}}}{(1-\gamma)^2} (\beta - \frac{1}{2}p_{\text{RA}} (1-\alpha))$ (details in Theorem 3) , indicating our method can bridge both the theoretical and practical performance of policies.
> > >
> > > **Q2: Is it possible to effectively incorporate uncertainty sets into offline datasets in a model-free context?**
> > >
> > > A2:
> > >
> > > In offline reinforcement learning (RL), our sole data source is the offline dataset. To develop a robust model that accounts for uncertainty in transition probabilities, traditional methods often rely on model-based approaches, such as the one proposed in MoREL (Kidambi et al. 2020). These methods aim to achieve robustness by explicitly modeling the transition probabilities. However, our paper introduces a novel approach that targets the same goal of transition probability robustness but eliminates the need to train a separate transition model. We achieve this by constructing an uncertainty set within a state-adversarial Markov Decision Process (MDP). This method allows us to prepare for uncertainties in transition probabilities using only the available offline dataset, without the additional complexity of model-based methods.

---

### Official Review · Reviewer_8FWn · 2023-11-01

**Soundness:** 1 poor
**Presentation:** 1 poor
**Contribution:** 2 fair
**Rating:** 5
**Confidence:** 4

**Summary:**

The paper introduces Relaxed State-Adversarial Offline Reinforcement Learning (RAORL), a model-free approach for offline RL. RAORL reframes the policy problem as a state-adversarial optimization challenge, eliminating the need for explicit environmental modeling and addressing model uncertainty issues. The method guarantees policy robustness and adaptability to dynamic transitions. Empirical evaluations on the D4RL benchmark demonstrate RAORL's superiority over baseline methods in continuous-control tasks, highlighting its potential for risk-sensitive applications.

**Strengths:**

1. The paper introduces an intriguing concept by integrating adversarial robust reinforcement learning with offline model-free reinforcement learning within a state-adversarial optimization framework.

2. The paper's solid theoretical foundation and validation of the reality gap contribute to its strength.

**Weaknesses:**

1. The paper's structure is not well-organized, making it challenging to grasp the primary contributions and follow the narrative. Furthermore, the writing is convoluted, containing multiple unclear sentences that hinder comprehension.

2. The experimental results lack persuasiveness. Firstly, the authors do not compare their work with state-of-the-art algorithms such as SAC-N, EDAC [1], and LB-SAC [2]. Even when implementing the algorithms based on https://github.com/tinkoff-ai/CORL, this paper's results are inferior to previous methods in most Mujoco and Adroit environments. Hence, I believe there are significant limitations in the paper's empirical contributions.

3. Figure 2 should clarify the color codes representing different algorithms and indicate the corresponding perturbation magnitudes. The paper should explain how optimal state perturbations were determined in the robustness analysis. The connection between the state perturbations in the experiments and the real-world environments, as claimed in the paper, seems tenuous. I have reservations about the practical significance of this paper.

4. While the paper focuses on model-free offline RL, I find it puzzling that model-based offline RL is included in the related work. Robust RL works like [3, 4] (and others) dealing with state adversaries seem more relevant, yet the authors do not discuss them in the related work section.

[1] Uncertainty-Based Offline Reinforcement Learning with Diversified Q-Ensemble. Gaon An, Seungyong Moon,  Jang-Hyun Kim, Hyun Oh Song. NeurIPS 2021.

[2] Q-Ensemble for Offline RL: Don’t Scale the Ensemble, Scale the Batch Size. Alexander Nikulin, Vladislav Kurenkov, Denis Tarasov, Dmitry Akimov, Sergey Kolesnikov.

[3] Robust Reinforcement Learning on State Observations with Learned Optimal Adversary. Huan Zhang, Hongge Chen, Duane Boning, Cho-Jui Hsieh. ICLR 2021.

[4] Efficient Adversarial Training without Attacking: Worst-Case-Aware Robust Reinforcement Learning. Yongyuan Liang, Yanchao Sun, Ruijie Zheng, Furong Huang. Neurips 2022.

**Questions:**

1. How does RAORL effectively bridge the performance gap in real-world applications, as claimed in the introduction?

2. How to determine the attack budget ($\epsilon$) in RAORL?

3. I am seeking clarity regarding the primary advantage of RAORL. Is it intended to be more effective than other offline model-free algorithms, aimed at improving adversarial robustness, or targeting enhanced offline performance? While the paper mentions these advantages, the results do not strongly demonstrate superior performance or robustness with enough evidence and explanation.

---

> ### Author Response · Authors · 2023-11-21
> **Rebuttal**
>
> We thank reviewer 8FWn for the insightful comments. Below, we address the questions raised by the reviewer. We hope the replies could help the reviewer further recognize our contributions. Thank you.
>
> > Q: The paper's structure is not well-organized, making it challenging to grasp the primary contributions and follow the narrative. Furthermore, the writing is convoluted, containing multiple unclear sentences that hinder comprehension.
>
> A: In the second paragraph of Section 4, we provided a logical thread of sections of 4.1, 4.2 and 4.3. In Section 4.1, we systematically measure the performance discrepancies between the policy shaped solely from the offline dataset (P_B) and its application in the universal set of transition kernels (U). Sections 4.2 and 4.3 further build upon this by demonstrating methods to enhance policy performance.
>
>
> > Q: The experimental results lack persuasiveness. Firstly, the authors do not compare their work with state-of-the-art algorithms such as SAC-N, EDAC [1], and LB-SAC [2]. Even when implementing the algorithms based on https://github.com/tinkoff-ai/CORL, this paper's results are inferior to previous methods in most Mujoco and Adroit environments. Hence, I believe there are significant limitations in the paper's empirical contributions.
>
> A: SAC-N, EDAC, and LB-SAC are all ensemble-based algorithms where the ensemble sizes are 500, 50, and 50, respectively. These ensemble-based algorithms often require extensive computational resources due to their reliance on large ensembles for uncertainty estimation (Nikulin et al., 2022). We did not include these algorithms in the comparative study because our method is not only model-free but also ensemble-free.
>
> > Q: Figure 2 should clarify the color codes representing different algorithms and indicate the corresponding perturbation magnitudes. The paper should explain how optimal state perturbations were determined in the robustness analysis. The connection between the state perturbations in the experiments and the real-world environments, as claimed in the paper, seems tenuous. I have reservations about the practical significance of this paper.
>
> A: In the caption of Figure 2, we have clarified “The blue and red solid lines depict the average performances of RAORL and ReBrac, respectively, in the presence of state perturbations”.
>
> In offline reinforcement learning (RL), a common and practical challenge arises when data collected from one system (Machine A) is used to train an agent that will be deployed on a different but similar system (Machine B). Even minor differences between these two machines can lead to distinct Markov Decision Processes (MDPs), posing a significant challenge in terms of MDP generalization. This situation underscores the importance of developing RL agents that can generalize effectively across varying MDPs. In essence, the agent must be capable of adapting to the nuances and potential discrepancies between the training environment (Machine A) and the deployment environment (Machine B).  The experiments in Figure 2 simulate this situation.
>
> > Q: While the paper focuses on model-free offline RL, I find it puzzling that model-based offline RL is included in the related work. Robust RL works like [3, 4] (and others) dealing with state adversaries seem more relevant, yet the authors do not discuss them in the related work section.
>
> A: The inclusion of model-based offline RL in the related work is to present a comprehensive background, as it also addresses offline scenarios. We acknowledge the relevance of robust RL literature, particularly works that handle state adversaries. We will incorporate references [3, 4] into our discussion on robust reinforcement learning in Section 2.3 to provide a more complete overview of the field and its developments.
>
> > Q: How does RAORL effectively bridge the performance gap in real-world applications, as claimed in the introduction?
>
> A: RAORL utilizes a strategic training approach that involves the uncertainty set of an offline dataset, as defined in Definition 4 of our paper. This approach ensures that RAORL is not confined to the narrow bounds of the data it was trained on but instead has exposure to a wider array of potential scenarios within its uncertainty set. Moreover, Theorem 3 in our paper underpins this methodology with a solid theoretical foundation, demonstrating how training within this uncertainty set can reduce the discrepancy between the policy's performance in the training data and its expected performance in real-world conditions.
>
> > Q: How to determine the attack budget () in RAORL?
>
> A: We set perturbation radius from the set {0.03, 0.05, 0.08, 0.1} multiplied by state differences (in Section 4.4). The setting of attack budgets were chosen to reflect the mean magnitude of the state affected by actions taken in each environment.

---

> ### Author Response · Authors · 2023-11-21
> **Rebuttal**
>
> > Q: I am seeking clarity regarding the primary advantage of RAORL. Is it intended to be more effective than other offline model-free algorithms, aimed at improving adversarial robustness, or targeting enhanced offline performance? While the paper mentions these advantages, the results do not strongly demonstrate superior performance or robustness with enough evidence and explanation.
>
> A: The primary objective of RAORL is to offer an effective and efficient solution in the realm of offline model-free algorithms.  It significantly reduces the computational and memory overhead typically associated with ensemble methods. In terms of performance and robustness, RAORL is designed to be competitive with other state-of-the-art offline model-free algorithms.

---

> > ### Comment · Reviewer_8FWn · 2023-11-22
> > **Thanks to the response**
> >
> > I appreciate the response from the authors, which has led to improvements in the presentations of this paper. While I still have some concerns regarding the motivation and novelty, I have decided to raise my score.

---

### Official Review · Reviewer_YpUx · 2023-11-01

**Soundness:** 3 good
**Presentation:** 2 fair
**Contribution:** 2 fair
**Rating:** 5
**Confidence:** 3

**Summary:**

The paper takes an adversarial approach to offline RL in order to account for the ambiguity in transition dynamics that arises from limited samples. The authors present theory outlining performance guarantees for a “risk-aware” policy which optimizes the average performance across an uncertainty set of plausible transition dynamics which have low TV distance to the empirical distribution. They propose to build this uncertainty set by constructing dynamics which are perturbed to be pessimistic with respect to the policy’s value function. The practical implementation of this pessimism can be achieved in continuous state spaces using the Fast Gradient Sign Method to compute adversarial perturbations. The resulting algorithm, Relaxed State-Adversarial Offline RL (RAORL), is evaluated on the D4RL benchmark, where it achieves strong results.

**Strengths:**

* The algorithm adapts a previous online robust RL algorithm, Relaxed State-Adversarial Policy Optimization (RAPPO), to the offline setting.
* The paper presents theoretical guarantees for an idealized version of the algorithm.
* The performance of RAORL on D4RL is strong compared to prior methods, and the state-adversarial training improves (or doesn’t hurt) performance compared to the base algorithm, ReBrac. The gain over ReBrac is most apparent on the AntMaze tasks.

**Weaknesses:**

* As far as I can tell, the RAORL algorithm is not substantially different from RAPPO, other than that (i) it is applied offline, (ii) a different solver is used (ReBrac vs. PPO), although this is necessitated by (i), and (iii) the relaxation parameter $\alpha$ is tuned as a hyperparameter rather than being updated as part of the algorithm. It would be helpful for the reader if you describe the distinction more clearly.
* Similarities between this paper’s theoretical results and those of the MoREL paper (Kidambi et al. 2020) are not highlighted. For example, Lemma 1 here looks nearly identical to Theorem 1 in that paper, with the addition of expectation over the uncertainty set $\mathcal{U}$. It would be helpful for the reader if you include some comments on the similarities/differences, perhaps in an appendix.
* Although it is repeatedly mentioned that one of the primary motivations for avoiding a model-based algorithm is that they struggle with stochastic environments, all the experiments are conducted on deterministic tasks.
* I felt that the self-promotional language should be toned down significantly in the writing of this paper (e.g. “A Leap Towards”, “innovative”, “top contender”, “superior”, “impressive”, etc.). I understand that some degree of salesmanship is standard in today’s research environment, but in my opinion it is more appropriate for Twitter than in the paper itself.

**Questions:**

* Am I missing other differences between RAORL and RAPPO?
* In Definition 1, is the expectation over $s,a$ taken over all transitions in the offline dataset? (I think the answer is yes, but wanted to confirm. The notation could be modified to clarify this.)
* In Definition 3, do the transition probabilities need to be re-normalized if the argmin defining $Z^\pi_\sigma$ contains more than one state? If we are assuming the argmin is always unique, perhaps this should be noted in the paper.
* The definition of $p_r$ (in Lemma 3 in Appendix A.2) is given as “the probabitlity [sic] of $P \in \mathcal{U}_r$ for every $P \in \mathcal{U}$”. I am not sure how to interpret this?
* Could you describe more precisely how the perturbations were chosen in section 5.3?

---

> ### Author Response · Authors · 2023-11-21
> **Rebuttal**
>
> We thank reviewer YpUx for the insightful comments. Below, we address the questions raised by the reviewer. We hope the replies could help the reviewer further recognize our contributions. Thank you.
>
>
> > Q: As far as I can tell, the RAORL algorithm is not substantially different from RAPPO, other than that (i) it is applied offline, (ii) a different solver is used (ReBrac vs. PPO), although this is necessitated by (i), and (iii) the relaxation parameter is tuned as a hyperparameter rather than being updated as part of the algorithm. It would be helpful for the reader if you describe the distinction more clearly.
>
> A: RAORL is tailored for offline RL, focusing on adaptability across multiple domains using static data, training within State-Adversarial uncertainty sets to ensure robust performance across a range of MDPs. Theorem 3 in our work theoretically validates the efficacy of RAORL in these offline scenarios. On the other hand, RAPPO is concerned with online settings, aiming to enhance both average and worst-case performance concurrently. These distinct goals highlight the fundamental differences in the application and contribution of RAORL compared to RAPPO.
>
>
> > Q: Similarities between this paper’s theoretical results and those of the MoREL paper (Kidambi et al. 2020) are not highlighted. For example, Lemma 1 here looks nearly identical to Theorem 1 in that paper, with the addition of expectation over the uncertainty set.  It would be helpful for the reader if you include some comments on the similarities/differences, perhaps in an appendix.
>
> A: Our Lemma 1 and MoREL's Theorem 1 differ primarily in their treatment of transition probabilities. MoREL assumes known transition probabilities, using them to create a pessimistic MDP. In Theorem 1, MoREL work with the constraint that the Total Variation Distance (DTV) between the model transition probability $\hat{P}(\cdot|s,a)$ and the true probability $P(\cdot|s,a)$ is less than or equal to $\alpha$. However, in practice, $P(\cdot|s,a)$ is unknown since we only have access to the transition kernel derived from the offline dataset. In contrast, RAORL operates without this assumption, reflecting real-world scenarios where true dynamics are often unknown.
>
>
> > Q: In Definition 1, is the expectation over taken over all transitions in the offline dataset? (I think the answer is yes, but wanted to confirm. The notation could be modified to clarify this.)
>
> A: In Definition 1, the expectation indeed goes beyond the transitions present in the offline dataset. It encompasses all possible transition kernels within a universal uncertainty set, representing potential dynamics in real-world environments. This approach extends our analysis beyond limited empirical data, preparing the model for a broader range of scenarios.
> However, it's important to note that while the concept of a universal uncertainty set might seem overly extensive, in practice, we refine this to a more manageable subset, denoted as $U_r$ in section 4.2 and $U_{\epsilon}$  in section 4.3. This refinement process involves constraining the uncertainty set based on realistic assumptions and characteristics observed in the offline data, thereby making the set both practical and relevant to real-world applications.
>
>
> > Q: In Definition 3, do the transition probabilities need to be re-normalized if the argmin defining contains more than one state? If we are assuming the argmin is always unique, perhaps this should be noted in the paper.
>
> A: The problem can be easily solved by setting a rule, such as picking the state that has the smallest value in the first dimension if there are multiple choices. In our implementation, the worst state is determined using the FGSM method. Although there can be multiple states that are equally worse, the FGSM finds only one of them. We will clarify this issue in the revision.
>
> > Q: The definition of  (in Lemma 3 in Appendix A.2) is given as “the probabitlity [sic] of  for every ∈”. I am not sure how to interpret this?
>
> A: In Lemma 3, $ p_r$ represents the likelihood of each possible transition kernel $ P $ being within a robust subset $U_r $, which is taken into account by the robust policy $\pi_r$.
>
> > Q: Could you describe more precisely how the perturbations were chosen in section 5.3?
>
> A: We defined the perturbation magnitude linearly with 20 points in the range $[0, 0.1]$. A perturbation of 0 signifies no perturbation, while a perturbation of 0.1 implies a perturbation strength of $0.1 \times | \text{ActualNextState} -  \text{State}|$, thereby ensuring that the norm of the difference between the actual state and the perturbed state is at most $0.1 \times  | \text{ActualNextState} -  \text{State}|$.
>
> As depicted in Figure 2, we analyzed the effect of different perturbation magnitudes on agent performance. The results demonstrate that RAORL maintains robust performance even under significant state perturbations.

---

> > ### Comment · Reviewer_YpUx · 2023-11-22
> > **Response to authors**
> >
> > Thank you for your response. My questions are mostly answered. However my first question was more asking about the *algorithmic* differences, rather than differences in setting. Perhaps more directly, how is RAORL different from RAPPO if you were to apply both to a static offline dataset?
> >
> > Also, I disagree that "MoREL assumes known transition probabilities". It is estimating the dynamics from data. The divergence between model probabilities and true probabilities can be bounded using standard concentration inequalities.

---

> > > ### Author Response · Authors · 2023-11-23
> > > **Response to Reviewer YpUx**
> > >
> > > We thank the reviewer for the questions!
> > >
> > > The implementation of RAORL is similar to RAPPO, yet the two approaches diverge fundamentally in their theoretical perspectives. Our research establishes a theoretical bound on the gap between offline datasets and real-world scenarios, substantiating the assertion that introducing state adversaries significantly bolsters the resilience of policies derived from static datasets. To the best of our understanding, RAORL stands out as a pioneering method in its integration of uncertainty sets within the framework of solving offline RL challenges. Relaxed State-Adversarial method (RAPPO) is only a step utilized to aggregate uncertain transitions when training policies. **Specifically, the goal of our study is to address two pivotal questions:**
> > >
> > >
> > > **Q1: To what extent and through what mechanisms can we reduce the performance gap between offline datasets and real-world scenarios?**
> > >
> > > A1:
> > >
> > > Our primary aim is to lessen the performance gap typically observed when a policy developed using offline datasets (denoted as $P_B$) is applied in real-world scenarios, characterized by a diverse set of transition kernels (represented by $U$). We approach this in two key ways:
> > >
> > > 1. **Risk-Aware Policy Training:** We demonstrate that training a risk-aware policy over a carefully constructed uncertainty set $U_r$ can reduce the gap between its performance in training and real-world applications. This reduction is quantified by: $\frac{4 \gamma R_{\text{max}}}{(1-\gamma)^2} (\beta - \frac{1}{2}p_r \beta_r)$ (details in Theorem 2).This formula represents a theoretical measure of the improvement we can expect when using this risk-aware approach.
> > >
> > > 2. **Constructing a Practical Uncertainty Set:** We create an uncertainty set, $U_r$, using a state-adversarial uncertainty set  $U_\epsilon^{\pi}$ as outlined in Definition 4. The theoretical benefit of this approach is captured by $\frac{4 \gamma R_{\text{max}}}{(1-\gamma)^2} (\beta - \frac{1}{2}p_{\text{RA}} (1-\alpha))$ (details in Theorem 3) , indicating our method can bridge both the theoretical and practical performance of policies.
> > >
> > > **Q2: Is it possible to effectively incorporate uncertainty sets into offline datasets in a model-free context?**
> > >
> > > A2:
> > >
> > > In offline reinforcement learning (RL), our sole data source is the offline dataset. To develop a robust model that accounts for uncertainty in transition probabilities, traditional methods often rely on model-based approaches, such as the one proposed in MoREL (Kidambi et al. 2020). These methods aim to achieve robustness by explicitly modeling the transition probabilities. However, our paper introduces a novel approach that targets the same goal of transition probability robustness but eliminates the need to train a separate transition model. We achieve this by constructing an uncertainty set within a state-adversarial Markov Decision Process (MDP). This method allows us to prepare for uncertainties in transition probabilities using only the available offline dataset, without the additional complexity of model-based methods.

---

### Meta-Review · Area_Chair_udkp · 2023-12-09

**Metareview:**

The paper proposes Relaxed State-Adversarial Offline Reinforcement Learning (RAORL), a new approach for offline RL that has potential in risk-sensitive applications. The state-adversarial formulation allows RAORL  to tackle model uncertainty issues. However, the reviewers have raised several concerns including limited comparisons to state-of-the-art offline RL algorithms and a confusing presentation / writing. Both issues have been raised by multiple reviewers, and the rebuttal from the authors have not sufficiently convinced them.

**Justification For Why Not Higher Score:**

Convoluted writing / presentation combined with limited experimental significance compared to state-of-the-art offline RL algorithms.

**Justification For Why Not Lower Score:**

N/A

---

### Decision · Program_Chairs · 2024-01-16

Reject